Positive emotion induction improves cardiovascular coping with a cognitive task

Molins Francisco
Pérez-Calleja Tania
Abad-Tortosa Diana
Alacreu-Crespo Adrian
Serrano-Rosa Miguel Ángel m.angel.serrano@uv.es
Department of Psychobiology, Universidad de Valencia , Valencia , Spain
Scorolli Claudia
Electronic publication date: 2021 Mar 12
Publication date: 2021
Volume: 9
Electronic Location ID: e10904
Received 2020 Sep 18; Accepted 2021 Jan 14
Copyright: ©2021 Molins et al.
Copyright year: 2021
Copyright holder: Molins et al.
License: This is an open access article distributed under the terms of the Creative Commons Attribution License, which permits unrestricted use, distribution, reproduction and adaptation in any medium and for any purpose provided that it is properly attributed. For attribution, the original author(s), title, publication source (PeerJ) and either DOI or URL of the article must be cited.
License URL: https://creativecommons.org/licenses/by/4.0/

Keywords: Positive emotions, Emotional induction, Tower of Hanoi, Heart-rate variability, Cognition, Coping

Funding: The authors received no funding for this work.

==============================
Feeling positive emotions seems to favour an adaptive cardiovascular response (greater heart rate variability, HRV), associated with improved cognitive performance. This study aims to test whether the induction of a positive emotional state produce such cardiovascular response and therefore, enhance coping and performance in Tower of Hanoi (ToH). Forty-two Participants were randomly distributed into two groups (Experimental and Control). Experimental group was subjected to the evocation of a memory of success, while control group was subjected to an attentional task before performing ToH. Heart Rate Variability (HRV), activity of the zygomatic major muscle (ZEMG) and emotions were measured. Emotional induction increased ZEMG activity, feelings of emotional valence and HRV, but the performance in ToH was not different from control. Experiencing positive emotions seems to favour an adaptive psychophysiological response when faced with a complex cognitive task. These results are discussed in relation to clinical practice and health.

Introduction

Although there is a certain stable predisposition in our perceptions, cognitive capacities, and behavior, all of them can be influenced by the punctual state of the organism. This is the premise of the embodied brain approach (Barrett, 2011; Kiverstein & Miller, 2015), which highlights the need to study the brain in interaction with the body to really understand cognition and behavior. Classic hierarchical models (e.g., the mammalian brain; MacLean, 1990) maintain a “corticocentric” point of view (Kiverstein & Miller, 2015) where the development of the brain cortex would be the one that confers superior cognitive abilities. However, from the embodied brain, it is discussed a reciprocal relationship between the cortex and subcortical regions (including the brainstem and cerebellum), the latter being necessary for the rise of cognitive abilities and, at the same time, modulating their functioning (Parvizi, 2009). Therefore, this approach understands that cognition and emotions are inextricably linked (Mueller, 2011; Pessoa, 2013; Pessoa, 2008). Emotions, seen as “states of action readiness manifest in the body as forms of arousal” (Kiverstein & Miller, 2015, p. 9), are related to the homeostatic equilibrium of the organism in interaction with the environment (Colombetti, 2014), and modulate cognition due to the feedback they provide on the neocortex (Colombetti, 2014; Mueller, 2011; Pessoa, 2013).

Maybe due to traditional “corticocentric myopia” (Kiverstein & Miller, 2015), it is common to find in cognitive sciences a negative connotation when talking about emotions, considering that these can bias cognition and that repressing them is beneficial (Kahneman, 2011; Li et al., 2017; Sunstein, 2014). It is true that negative emotions such as anger or fear favor defensive or orientation responses (Bradley, 2009), guiding attention towards the most relevant stimuli in the environment (Barros et al., 2017) and, consequently, limiting some cognitive capacities such as executive functions, working memory or the ability to regulate emotions (Shook, Fazio & Vasey, 2007; Shook et al., 2007; Thayer & Friedman, 2004). However, it would be wrong to interpret this as negative per se. In fact, from an evolutionary perspective it would have an adaptive sense, maximizing survival by ignoring less relevant information and providing fast responses to threatening stimuli (Barros et al., 2017; Thayer, Julian & Lane, 2009; Vuilleumier, 2005). On the other hand, regarding the impact of positive emotions on cognition, it has been observed that experiencing emotions such as joy, surprise or happiness promotes mental states that help to solve problems and achieve goals (Fredrickson, 2001; Stuss & Levine, 2002). Concretely, positive emotions would favor the working memory (Gray, 2001), verbal fluency (Carvalho & Ready, 2010), taking perspective on a problem (Herrington et al., 2005), information processing and the ability to connect thoughts and ideas in an integrated and flexible way (Fredrickson, 1998; Fredrickson & Losada, 2005; Herrington et al., 2005), among other cognitive skills (Mueller, 2011).

As decades of research highlighted, prefrontal cortex (PFC) plays a central role in the functioning of the cognitive skills mentioned above (Carvalho & Ready, 2010; Herrington et al., 2005; Mueller, 2011). For example, imaging data indicated greater PFC activation during a phonemic fluency task (Billingsley et al., 2004), or during the relevant-information selection in an attentional task (Milham et al., 2003), among others. And substantial clinical data evidenced that individuals with deficits in the PFC, especially in the dorsolateral region (DLPFC), experiment difficulties when anticipating future events or choosing among multiple problem-solving strategies (for a review, see Banich, 2004). The specific mechanisms by which positive emotions could improve cognitive skills are still unknown, but it seems that “they are processes that share neuroanatomical underpinnings” (Carvalho & Ready, 2010, p.228). So, neuroscience has indicated that positive emotions also activate PFC, specifically DLPFC (Gray, Braver & Raichle, 2002; Herrington et al., 2005), with a predominant lateralization of the left hemisphere, widely described by electroencephalogram (EEG) (Carvalho & Ready, 2010; Davidson, 2004; Heller & Nitschke, 1998) and functional magnetic resonance (fMRI) (Gray, Braver & Raichle, 2002; Herrington et al., 2005). Thus, as Carvalho & Ready (2010) suggested, although more research is needed to clarify the specific mechanisms, the relationship between positive emotions and the higher cognitive performance could be due to the activation of overlapping cortical areas. In this line, Andreau & Torres Batán (2019) found that selective pre-activation of brain circuits participating in the verbal stimuli recognition could favor the memory of words. And in fact, the own Transcranial direct current stimulation (TDCS) method tries to pre-activate brain regions to enhance certain cognitive functions, such as language (Meinzer et al., 2014).

As Brugnera et al. (2018) have recently begun to point out, another way of studying the relationship between positive emotions and cognition, in addition to neuroimaging, would be attending to heart rate variability (HRV). Firstly, experiencing positive emotions, unlike negative ones, does not seem to produce a great withdrawal of the parasympathetic nervous system (PNS) or an increase of the sympathetic nervous system (SNS); in fact, it would contribute to improve the sympathetic-parasympathetic balance (Brugnera et al., 2018). Therefore, through the influence that both autonomous branches exert on the heart, there would be an increase in the HRV or, at least, it would not decrease so much (Brugnera et al., 2018; Shaffer & Ginsberg, 2017) as usually happens when facing a challenge or stressor in a neutral emotional state (Castaldo et al., 2015). On the other hand, in line with the Neurovisceral Integration Model (NIM; Park & Thayer, 2014; Thayer & Friedman, 2004; Thayer, Julian & Lane, 2009), the increased activity of PFC (induced in this case by positive emotions) would restrain impulses from subcortical structures and would produce a reflex through the vagus nerve which would also be reflected in the increase of HRV (Park & Thayer, 2014). Therefore, in both ways, inducing a positive emotion could increase HRV, or avoid a large reduction in HRV in the face of a challenge. A high HRV, at rest and phasic (i.e., in response to a stressor), is considered an adaptive cardiovascular activity, as it is related to greater cognitive flexibility, better emotional regulation and greater performance in complex tasks that depend on executive functions (Grossmann, Sahdra & Ciarrochi, 2016; McCraty & Shaffer, 2015; Park & Thayer, 2014; Shook et al., 2007; Thayer, Julian & Lane, 2009). Thus, although the specificity of HRV would not be comparable and could not replace neuroimaging techniques, this could also be a useful, economic, and non-invasive method for study how positive emotions affect cognition.

Therefore, the objective of this study is to test whether the induction of a positive emotional state could produce such adaptive cardiovascular response, associated with higher cognitive performance and therefore, in line with Isen (2000), whether this benefits coping and performance in a task that depends on complex cognitive skills, such as the Tower of Hanoi (ToH; Barroso y Martín & León-Carrión, 2001). The execution of ToH has been linked to the capacity of inhibition, planning, logical reasoning, cognitive flexibility, working memory and problem solving (Barroso y Martín & León-Carrión, 2001; Goldstein et al., 2014), so the strengthening of these skills could improve performance in this task. This performance is usually measured based on the time and number of movements dedicated to solving ToH, as well as, within a fixed time, the number of towers that a person can solve (Barroso y Martín & León-Carrión, 2001). Thus, the higher the number of towers solved, as well as the lower the number of movements and time to solve each tower, the greater the performance.

Regarding the induction of the positive emotional state, classical methods such as reading and internalizing positive self-affirmations, seeing happy facial expressions, or listening to music, have proven effective (Gerrards-Hesse, Spies & Hesse, 1994). However, one of the most recent methods used is the evocation of autobiographical memories about successful life experiences (Brugnera et al., 2018; Yousefi et al., 2015). This method is based on the reminiscence therapy and has proven to be useful in increasing self-esteem, satisfaction with life and happiness (Yousefi et al., 2015). In this study we decided to use this method to carry out positive emotional induction (see ‘Material & Methods’ for more details). In order to ensure that an emotional change really occurs after induction, it can be evaluated subjectively using questionnaires such as Self-Assessment Manikin (SAM; Bradley & Lang, 1994), but also, since the activity of the zygomatic major muscle has been related to the experience of positive emotions (Cacioppo et al., 2000; Cacioppo & Fridlund, 1986; Ravaja et al., 2006; Schmidt et al., 2006), its measurement would constitute a good objective indicator of the success of emotional induction. Thus, our hypotheses posit that (hypothesis 1) inducing a positive emotional state in some participants will increase the activity of their zygomatic major muscle and their positive emotional valence measured with SAM; (hypothesis 2) this will cause them a higher HRV during the performance of ToH; (hypothesis 3) and this performance will be higher: they will solve more towers in less time and number of movements; all this with respect to a control group not exposed to induction.

Materials & Methods

Participants

Based on the effect sizes found in previous works on emotions induction and HRV (η2p = .30; Brugnera et al., 2018), as well as on emotions induction and cognitive performance (η2p = .16; Gray, Braver & Raichle, 2002), a priori power analyses using G*Power indicated a requisite of 30–48 participants (power = 80%, α = .05) to perform ANOVAS comparing between groups (intervention vs control) their performance in ToH and their HRV. We recruited 50 participants, however, 8 were eliminated due to drug consumption and technical issues. A total of 42 healthy volunteers (age: M = 23, SD = 2.06; women: N = 28;) were finally included. All of them, students from the University of Valencia, recruited in the classes by asking them if they wanted to participate in exchange for academic credits. Those interested filled out a self-administered questionnaire to ensure that they met the following inclusion criteria: having between 18 and 30 years old; not having cardiovascular, endocrine, neurological or psychiatric diseases; not consuming more than 10 cigarettes per day; not consuming drugs and alcohol habitually; not doing more than 10 h of physical exercise per week and not having experienced a highly stressful event in the last month. In addition, variables, such as not having carried out strenuous exercise, consumed drugs or alcohol in the last 24 h, nor smoked or drank stimulant drinks in the 2 h prior to the experiment were controlled upon arrival at the laboratory. The participants were randomly distributed in two groups, experimental (N = 21) and control (N = 21) and both groups were checked for homogeneity in age, gender proportion, body mass index, emotional state through the Self-assessment manikin (SAM), and emotional traits through the Beck-II Depression Inventory (BDI-II) and the Anxiety Trait Questionnaire (STAI-R).

Questionnaires

Self-Assessment Manikin (SAM; Bradley & Lang, 1994). Evaluates by means of pictograms the three main emotional dimensions: valence, activation, and dominance. Each dimension goes from 1, the extreme of maximum pleasure (valence), greater activation (activation) and feeling of submission (dominance); up to 5, the extreme of maximum displeasure, relaxation and feeling of dominance, respectively.

Sociodemographic. Collects information on age, gender and other variables not included in this study.

Beck-II Depression Inventory (BDI-II; Beck, Steer & Brown, 1996). 21 items on Likert scale of 4 points (from 0 to 3), which must be added to evaluate depression. The higher the score, the greater the depression (α = .87).

Anxiety Trait Questionnaire (STAI-R; Spielberger, Gorsuch & Lushene, 1970). 20 items in Likert scale of 4 points (where 0 is “never” and 3 is “almost always”), which must be added to evaluate trait anxiety. The higher the score, the greater the anxiety (α = .85).

Tower of Hanoi (ToH)

The ToH consists of a platform with three parallel posts and a certain number of initial pieces, stacked downwards on the left post (the largest at the base, to the smallest at the cusp).The participants have to move all the pieces to the right post, maintaining the same descending order, but complying with the following rules: no more than one piece can be moved at a time, no one piece can be stored outside the posts, and no piece can be stacked on top of a smaller one. When all the pieces are correctly in the right post, the tower is completed; after this, and if there is still time, the task is restarted increasing the difficulty by adding one more piece to the left post. Participants have to solve as many towers as possible within a 6-minutes time limit. Before starting the task, a practice was carried out with 2 pieces. The actual execution started with 3 pieces and could increase up to a maximum of 7, depending on whether the previous towers were resolved. Three indicators were obtained to evaluate performance. The first one was the number of towers completed in 6 min (the more towers the better the performance). On the other hand, following Barroso y Martín & León-Carrión (2001), we focused on those towers that the whole sample managed to complete within the time limit (in this case, only the towers with 3 and 4 pieces were completed by all participants). For each participant, the number of movements and the time needed to complete each tower individually were counted. Thus, the fewer the movements and time in each tower, the better the performance in that tower.

Intervention

In the experimental group, a positive emotional induction was carried out through the guided evocation of a successful autobiographical memory, based on reminiscence therapy, since this method has proven effective in increasing happiness (Yousefi et al., 2015). The induction was carried out by the same investigator for all participants. The guide (see annex 1) is composed of direct indications (e.g., “concentrate on the specific place where it occurred”) and questions (e.g., “ how was that place?”), techniques drawn from the imagined method of practice (Palmi & Mariné, 1996) and reminiscence therapy (Yousefi et al., 2015), respectively. For its part, the control group listened to a neutral text (inspired by a construction documentary; see annex 2). They were instructed to pay attention since at the end they would be asked about the text. The duration was also 10 min. In both cases, to ensure that both groups received the same number of sentences and the cadence of the text/stimulation, we controlled the number of sentences and the time of the pauses. The investigator knew to which group the participants had been assigned, but they did not know to which condition (intervention vs. control) they belonged. In addition, another researcher was responsible for analyzing the data without knowing the identity of the groups.

Physiological variables

The PowerLab/16SP (ADInstruments) equipment linked to the LabChart 5.2 software was used. The signals were transmitted at 1,000 Hz. The details of each signal are specified below.

Facial electromyography of the zygomatic muscle (ZEMG)

The average zygomatic muscle activity was obtained during the last 5 min of habituation (baseline, BL), after the ToH-1 and during the intervention time. The recording was made by placing three low voltage Ag/AgCl cup electrodes, one on the nose and the other two on the zygomatic major muscle, following Van Boxtel (2010) indications. A band-pass filter was established between 20 and 250 Hz and a 50 Hz high-pass filter.

Heart rate variability (HRV)

Three single-use Ag/AgCl cup electrodes with Einthoven’s third derivation were used to record the electrocardiogram and, in particular, the time interval between beats (R-R) allowing analysis of HRV. Following the recommendations of the Task Force (1996), we standardized 5-minute records so that they could be compared among them. Thus, the 5 min of BL and the 5 central of each ToH, were exported and analyzed with the free software Kubios HRV (Tarvainen et al., 2014). The signal was filtered using the smooth priors’ method and applying the level of artifact correction that was necessary for each case. Of all the indicators that can be analyzed within the HRV, the standard deviation of the interval between normal beats (SDNN) was used, i.e., those that remain after removing ectopic beats. The reasons for selecting this indicator are as follows: on the one hand, SDNN is among the main ones highlighted by the Task Force (1996) to study global HRV; this is because SDNN is mathematically identical to the total power of the spectral analysis or the sum of the Very-Low, Low and High frequency bands that can be extracted from the time domain, for example, by means of Fast Fourier Transformation (Task Force, 1996); Thus, it is an indicator of HRV that includes information of all the contributing cyclic components in their variation (Shaffer & Ginsberg, 2017; Task Force, 1996), which allows to speak about the increase or decrease of HRV in general, as is pretended in this study. Thus, a higher SDNN would imply a higher HRV. On the other hand, this minimizes the chance of α-inflation due to the multiple statistical contrasts avoided by not using the rest of the indicators (Grossmann, Sahdra & Ciarrochi, 2016), many of which also have mathematical redundancy and would only complicate the explanation (Brugnera et al., 2018). Finally, it appears that for short measurements (e.g., 5 min), the main contribution to SDNN variations comes from the respiratory sinus arrhythmia (RSA), controlled by the PNS (Shaffer & Ginsberg, 2017), thus, with a single indicator we could speak of global HRV, but also partially infer the behavior of the autonomic nervous system. As Shaffer & Ginsberg (2017) point out, this inference should only be made under normal breathing conditions (between 9 and 26 cycles per minute), however, since the participants were controlled for no physical activity and were at rest, following the recommendations of Denver, Reed & Porges (2007), the respiratory rate was not controlled.

Protocol

The study consists of a single session, the duration of which is approximately one hour and a half. All the participants took part in the session between 15:30 pm and 19:00 pm. Participants were asked to wait in the University hall and accompanied in an elevator to the laboratory, avoiding the use of stairs. The general procedure was reported, and written informed consent was obtained from the participants. Afterwards, they were connected to the polygraph and had 10 min of rest as habituation. They filled out the SAM questionnaire to assess their subjective emotional state and performed the first ToH (ToH-1) for 6 min. After this, the experimental group completed a second SAM and underwent positive emotional induction; the control group performed a distracting task (listening to a text with neutral content). After the intervention, they refilled the SAM and performed the second ToH (ToH-2). When they finished, they were disconnected from the electrodes, completed a battery of questionnaires (sociodemographic variables and emotional traits, to study the homogeneity of the groups) and were measured height and weight. The protocol was approved by the Research Ethics Committee of the University of Valencia (H1393232860606) in accordance with the ethical standards of the Declaration of Helsinki of 1969.

Statistical analyses

In order to detect and eliminate outliers in all the variables of interest the method of the 2.5 standard deviations and, for variables measured more than once, the Mahalanobis distance with the criterion of p < .001 were employed. Kolmogorov-Smirnoff with Lilliefors correction was used to check normality. The variables of SAM, ZEMG, HRV and time spent to complete the tower of 4 pieces in ToH-2, did not follow the normal distribution. This last variable, as well as those of HRV, were normalized using the log10 method (Field, 2009). For the remaining ones, non-parametric tests were used. Using One-way ANOVAs, the means of the experimental and control group were contrasted in trait questionnaires, performance in ToH (1 and 2) and HRV (in BL, ToH-1 and ToH-2). The Man-Whitney test is used for non-parametric contrasts with SAM and ZEMG in BL, after ToH-1 and after the intervention (or during, for ZEMG). The alpha significance level was set at .05. In addition, the ANOVAs have indicated the partial square eta (η2p) symbolizing the size of the effect and β-1 (power) symbolizing the power. In the Mann–Whitney test the effect size (r) was calculated with the formula: r=Z∕N. All analyses were performed with IBM SPSS Statistics 23.

Results

Preliminary analyses

As can be seen in Table 1, the distribution of the sample in experimental and control groups is homogeneous, with no statistically significant differences in age, BMI, BDI-II or STAI-R. In addition, although there are more women than men, the distribution of both genders in the two groups maintains a similar proportion.

ToH 1 and 2 results

Below are the results of the ANOVAs for each of the measures taken to evaluate the performance in ToH 1 and 2: the number of towers reached in 6 min and the number of movements and time used to overcome the towers of 3 and 4 pieces (see Table 2).

As we can see, during ToH-1, there are no statistically significant differences between groups for any of the analyzed measures. On the other hand, after the intervention, in ToH-2, there are no differences between groups, with the exception of the time spent solving the tower with 4 pieces (T4), the time spent by the experimental group being greater than that of the control.

SAM results

In Table 3 we can see the results of the Mann–Whitney test for the contrast between control and experimental group in each SAM dimensions (valence, activation and dominance), for BL, after the first ToH and after the intervention.

We can see that none of the emotional dimensions differ in a statistically significant way between experimental group and control in BL, neither after performing the first ToH. However, after the intervention, we can observe how the experimental group shows, with respect to the control group, a lower score in emotional valence, i.e., a more positive emotional valence.

ZEMG results

As can be seen in Table 4, in relation to the facial electromyography of the zygomatic major muscle, the Mann–Whitney test was performed to contrast the level of activity of the experimental group with respect to the control in BL, after ToH-1 and during the intervention. Analyses show that during BL and after ToH-1 there were no statistically significant differences between groups. However, during the intervention there were significant differences in favor of the experimental group, which showed a higher level of activity in comparison with the control group.

Table 1 Data comparison between groups.

Homogeneity between groups.

	Experimental (N = 21)	Control (N = 21)	F	gl between	gl intra	p-value	
Age	M = 22.9 ± 1.81	M = 23.1 ± 2.3	0.08	1	40	.76	
Sex							
Men	28.6%	38.1%	0.28a	1	14a	.59	
Women	71.4%	61.9%	0.14a	1	28a	.7	
BMI	M = 23.73 ± 5.39	M = 23.87 ± 3.91	0.01	1	40	.92	
BDI-II	M = 7.38 ± 6.34	M = 8.62 ± 6.32	0.4	1	40	.53	
STAI-R	M = 21.19 ± 9.93	M = 17.14 ± 7.35	2.25	1	40	.14	
Notes.

M mean

± SD

BMI weight(kg) / size(m)2

BDI-II sum of 21 items measuring depression

STAI-R sum of 20 items measuring anxiety trait

p- values were calculated using ANOVA (continuous variables) or chi-square (categorical variables).

a These values correspond to χ2 and not to F, as well as N and not a gl intra.

Table 2 Comparison between groups in Tower of Hanoi.

ANOVAs of performance indicators in ToH-1 and ToH-2.

		Experimental	Control	F	gl between	gl intra	p-value	η2p	β-1	
ToH-1	Completed towers	M = 2.3 ± 0.57	M = 2.35 ± 0.74	0.57	1	38	.81	.001	.05	
Movements T3	M = 7.22 ± 0.94	M = 7.24 ± 0.76	0.03	1	37	.95	0	.05	
Time T3	M = 16.42 ± 7.57	M = 13.9 ± 7.58	1.07	1	37	.3	.02	.17	
Movements T4	M = 33.95 ± 14.19	M = 27.58 ± 12.23	2.19	1	36	.14	.05	.3	
Time T4	M = 128.2 ± 83.96	M = 83.65 ± 76.02	3.09	1	38	.08	.07	.4	
ToH-2	Completed towers	M = 2.8 ± 0.41	M = 2.67 ± 0.48	0.9	1	39	.34	.9	.15	
Movements T3	M = 7.2 ± 0.89	M = 7.45 ± 1.14	0.59	1	38	.44	.01	.11	
Time T3	M = 11.26 ± 2.4	M = 11.55 ± 3.5	0.88	1	37	.76	.002	.06	
Movements T4	M = 29.5 ± 7.81	M = 24.57 ± 11.1	2.67	1	39	.11	.06	.35	
Time T4	M = 62.32 ± 24.57	M = 45.24 ± 24.32	7.44	1	38	.01**	.16	.75	
Notes.

ToH Tower of Hanoi

T3 tower with 3 pieces

T4 tower with 4 pieces

M mean

± SD

Although some contrasts were made with variables transformed with log10, the original means are shown to facilitate interpretation; p-values were calculated using ANOVA; η2p, effect size; β-1, power.

** Significant contrast to .01 level.

Table 3 Evolution of emotions during protocol in both groups (control and experimental).

Mann-Whitney Test for SAM emotional dimensions in BL, after ToH-1 and after intervention.

		Experimental (N = 21)	Control (N = 21)	U	p-value	r	
BL	Valence	Mdn = 3	Mdn = 2	165.5	.21	.18	
Activation	Mdn = 4	Mdn = 4	207.5	.71	.05	
Dominance	Mdn = 3	Mdn = 3	196.6	.7	.05	
Post ToH-1	Valence	Mdn = 3	Mdn = 3	208	.95	.008	
Activation	Mdn = 4	Mdn = 4	183.5	.27	.16	
Dominance	Mdn = 4	Mdn = 3	188.5	.4	.12	
Post Inter	Valence	Mdn = 2	Mdn = 3	80.5	<.001**	.56	
Activation	Mdn = 4	Mdn = 3	196.5	.51	.09	
Dominance	Mdn = 3	Mdn = 3	178.5	.25	.17	
Notes.

SAM Self-Assessment Manikin

BL Baseline

Post ToH-1 after the first Tower of Hanoi

Post Inter after the intervention

Mdn median

p-values were calculated using Mann–Whitney Test; r, effect size (r=Z∕N).

** Significant contrast to .01 level.

Table 4 Mann-Whitney Test for ZEMG in BL, after ToH-1 and after intervention.

		Experimental (N = 21)	Control (N = 21)	U	p-value	r	
BL	ZEMG	Mdn = .11	Mdn = .12	183.5	.43	.17	
Post ToH-1	ZEMG	Mdn = .17	Mdn = .16	203	.63	.10	
Inter	ZEMG	Mdn = .32	Mdn = .29	132	.017*	.36	
Notes.

ZEMG electromyography of the zygomatic major muscle

BL Baseline

Post ToH-1 after the first Tower of Hanoi

Inter intervention

Mdn median

p-values were calculated using Mann–Whitney Test; r, effect size (r=Z∕N).

* Significant contrast to .05 level.

HRV results

ANOVAs were performed to study the differences in SDNN between experimental group and control during BL, ToH-1 and ToH-2. The analyses show (see Fig. 1) the absence of differences between groups in both BL (p = .95) and ToH-1 (p = .33); however, during ToH-2 (after intervention), the experimental group has a significantly higher level of SDNN (M = 40.85, SD = 15.81) than the control group (M = 30.42, SD = 11.55), F(1, 37) = 6.05, p = .019, η2p = .14, power = .66.

Figure 1 Experimental and control group HRV levels during the protocol.

HRV levels. SDNN levels are shown (in ms; M + SD), for the two groups (experimental and control), at moments 1: Baseline; 2: Tower of Hanoi 1; and 3: Tower of Hanoi 2. At this last moment, after having experienced emotional induction, the experimental group shows a significantly higher SDNN level than the control group.

Discussion

This study aimed to investigate effects produced by the induction of a positive emotion, through the evocation of an autobiographical memory of success, on the subsequent performance in the Tower of Hanoi (ToH), on the basis that positive emotions could enhance the cognitive skills needed to perform this task (Fredrickson & Losada, 2005; Herrington et al., 2005; Mueller, 2011). The results are controversial because, although the performance obtained by the experimental group (with emotional induction) did not appear to be different from that of the control group (without emotional induction), the first group showed a more adaptive cardiovascular response during the task, with higher HRV, which is usually associated with a better coping of the task (Grossmann, Sahdra & Ciarrochi, 2016; Park & Thayer, 2014; Thayer, Julian & Lane, 2009). This leads us to think that positive emotions may not necessarily produce great changes in the result, but they could facilitate better coping with difficulties. These issues will be discussed in more detail below.

First, it is important to point out that the distribution of the participants in both groups was homogeneous, presenting an emotional state (in SAM) and trait (in BDI-II and STAI-R), a zygomatic activity (ZEMG) and HRV (SDNN) similar at the beginning of the experiment. As can be seen in Fig. 1, during the first ToH, both groups showed a similar decrease of HRV with respect to BL. This seems logical if we consider that the participants were facing a complex task that could act as an acute cognitive stressor (Ciabattoni et al., 2017), which would promote an immediate parasympathetic withdrawal and increased sympathetic activity (Castaldo et al., 2015; Hidalgo, Pulopulos & Salvador, 2019). This would explain the decrease in SDNN, a global HRV indicator that is affected by the autonomous changes described above (McCraty & Shaffer, 2015; Shaffer & Ginsberg, 2017; Task Force, 1996). On the other hand, after the first ToH, both the subjective emotional state and the ZEMG remained undifferentiated between groups. In this way, it could be interpreted that both were exposed to intervention with a similar emotional state. However, during the intervention, the group exposed to the guided evocation of the autobiographical memory of success (experimental group) showed significantly higher ZEMG activity than the control group, which merely listened to a narration about constructions. If we consider that ZEMG was indicated as an objective indicator of the experience of positive emotions (Cacioppo et al., 2000; Cacioppo & Fridlund, 1986; Ravaja et al., 2006; Schmidt et al., 2006), we could affirm that, as intended, emotional induction was effective in provoking positive emotions. This interpretation would also be reinforced by the subjective perception of the participants themselves, who expressed in the SAM questionnaire that they felt happier after the intervention. All this would confirm our first hypothesis.

Under this differential mood, both groups performed the second ToH. However, their performance scarcely differed between them. They did not complete more towers than in the first attempt, although both finished the towers of 3 and 4 pieces with less movements and time, probably a phenomenon attributable to a learning effect on the task (Ahonniska et al., 2000; Welsh & Huizinga, 2005). The only difference between groups was observed in the time taken to complete the tower of 4 pieces, the experimental group being slower than the control. According to the usual ToH evaluation criteria, this would be a worse performance indicator (Barroso y Martín & León-Carrión, 2001), against the expected improvement after the intervention proposed in our third hypothesis. However, focusing on cardiovascular activity during the task, we saw that, although both groups continued showing lower levels of HRV than in BL, the experimental group faced the second ToH with a significantly higher level of HRV than the control group. Looking at Fig. 1 we can add that these differences were not due to an increase in HRV, but rather to the fact that their level was not as much reduced during the second ToH, as seems to happen in the control group. Given that for short measurements, the fluctuations in the indicator used, SDNN, would be explained more by the RSA controlled by the PNS (McCraty & Shaffer, 2015; Shaffer & Ginsberg, 2017), it could be inferred that the experimental group was perhaps manifesting a lower parasympathetic withdrawal during the second ToH thanks to its positive emotional state. This would be in line with the results described by Brugnera et al. (2018), who found a similar effect but with frequency indicators. On the other hand, returning to the NIM approaches (Park & Thayer, 2014; Thayer & Friedman, 2004), this response may also be evidencing the increased activity of neocortical regions that were expected to induce positive emotions (Brugnera et al., 2018; Herrington et al., 2005) which would favor a less impulsive coping by improving self-control capacity (Segerstrom et al., 2011) and emotional regulation (Grossmann, Sahdra & Ciarrochi, 2016). In fact, this could be an explanation of the longest time spent by the experimental group to overcome the tower of 4 pieces. The group did not perform worse in general terms, completing the same towers in 6 min than the control group and executing the same number of movements. Thus, perhaps taking longer in that phase does not imply a worse performance per se, but simply that they were thinking more about the movements before executing them. This would highlight the need to rethink whether the criteria for assessing cognitive task performance are adequate and/or sufficient, as is already being discussed in other cognitive domains such as decision-making (Chater et al., 2018). This debate emphasizes that many normative criteria used to judge whether a person “performs well” are not necessarily suited to the demands of the real world (Haksöz, Katsikopoulos & Gigerenzer, 2018; Volz & Gigerenzer, 2012) or are not the only aspects to consider. For example, importance is given to the outcome, but rarely attention is focused on how the situation is dealt with (Brugnach et al., 2011; Keys & Schwartz, 2007), that is, the strategy used or the physiological or psychological state in which it is carried out. This could have implications in clinical practice, where it might be useful to provide strategies to deal with problems in a healthy way and not only focus on achieving a certain result, which may depend on multiple contextual factors that are beyond the control of the patient and therapist. Thus, in line with our second hypothesis, and independently of the result obtained in ToH, it seems that experiencing a positive emotion promoted a better response to difficulties, presenting an adaptive cardiovascular activity that is not only related to better cognitive performance (Grossmann, Sahdra & Ciarrochi, 2016), but also to better coping with stress and threatening situations (Alacreu-Crespo et al., 2018); (Thayer, Julian & Lane, 2009) and better overall health (McCraty & Shaffer, 2015; Thayer, Julian & Lane, 2009). In fact, the opposite, i.e., a low level of HRV, is associated with inadequate response to stress or even chronic stress (McCraty & Shaffer, 2015; Park & Thayer, 2014); with psychopathological disorders such as major depression, generalized anxiety, panic or schizophrenia (Park & Thayer, 2014); with health problems such as hypertension, diabetes, obesity, arthritis and some types of cancer (Park & Thayer, 2014; Thayer & Friedman, 2004; Thayer, Julian & Lane, 2009); and increased risk of heart attack and death (Grossmann, Sahdra & Ciarrochi, 2016).

Our study is not exempt from limitations and results must be considered addressing them. So, the sample used is composed by healthy young people with a university level education, however, emotional induction could produce other effects in a sample that, for example, has a different sociocultural level or that presents a cognitive impairment associated with an advanced age or various pathologies, so it would be good to study different samples to provide greater ecological validity to the results. Similarly, no direct effects of induction on cognitive performance have been found. This could be due to a lack of power produced by the reduced sample size; but it should also be noted that only ToH was used to evaluate cognitive performance. ToH is linked to inhibition capacity, planning, logical reasoning, cognitive flexibility, working memory and problem solving (Barroso y Martín & León-Carrión, 2001; Goldstein et al., 2014), but perhaps using tasks that focus on other cognitive domains, the effects were different. It should also be noted that the researcher applying the intervention knew which group the participants belonged to (intervention vs. control). Although participants did not know this information, and another external researcher was responsible for carrying out the analyses without knowing the identity of the groups, the fact that the researcher who applied the intervention knew this information could affect results. Triple-blinded research should be carried out to replicate findings in the future. Moreover, although the distribution is homogeneous in both groups, the number of males is small, which makes it impossible to study differences by gender, something that would be interesting given that, as mentioned, ToH could constitute an acute cognitive stressor (Ciabattoni et al., 2017) and in previous literature it has been pointed out that the physiological response to stress is differential according to gender (Hidalgo, Pulopulos & Salvador, 2019). Finally, the results on HRV have been discussed in line with the NIM approach and based on the hypotheses raised in this study This could constitute a bias based on previous knowledge and expectations. Since no other variables have been included to investigate whether the HRV response really constitutes, as we suggested, a better coping with the cognitive task, we recommend caution on these conclusions.

Thus, further research is needed to address these issues and make our results robust.

Conclusions

In view of the results, we must return to the initial premise of the embodiment: the brain should not be studied in isolation. Our data show once again that emotions and cognition are related, and that experiencing positive emotions favors a physiological response that enhances cognition and promotes better coping with difficulties, although this does not always translate into a better result. Moreover, training in the induction of positive emotions could be a useful tool in clinical practice, preparing patients to better cope with stress and the difficulties of life in a healthy way.

Supplemental Information

Supplemental Information 1 Annex 1 and 2 (in English).

Annex 1: Transcription of the instructions given to the participants in the positive emotional induction group.

Annex 2: Transcription of the instructions given to the control group participants.

Click here for additional data file.

Supplemental Information 2 Annex 1 and 2 (in Spanish).

Anexo 1: Transcripción de las instrucciones dadas a los participantes del grupo de inducción emocional positiva.

Anexo 2: Transcripción de las instrucciones dadas a los participantes del grupo control.

Click here for additional data file.

Supplemental Information 3 SPSS data for all the participants with all the measures employed in the study

Participants are in columns and measures are in rows.

Click here for additional data file.

Additional Information and Declarations

Competing Interests

Author Contributions

Human Ethics

Data Availability

The authors declare there are no competing interests.

Francisco Molins analyzed the data, prepared figures and/or tables, authored or reviewed drafts of the paper, and approved the final draft.

Tania Pérez-Calleja performed the experiments, authored or reviewed drafts of the paper, and approved the final draft.

Diana Abad-Tortosa conceived and designed the experiments, performed the experiments, prepared figures and/or tables, and approved the final draft.

Adrian Alacreu-Crespo conceived and designed the experiments, authored or reviewed drafts of the paper, and approved the final draft.

Miguel Ángel Serrano-Rosa conceived and designed the experiments, analyzed the data, prepared figures and/or tables, authored or reviewed drafts of the paper, and approved the final draft.

The following information was supplied relating to ethical approvals (i.e., approving body and any reference numbers):

The Research Ethics Committee of the University of Valencia

approved the protocol of this study (H1393232860606).

The following information was supplied regarding data availability:

Raw data are available in the Supplemental Files.

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
