# Peer review of "Positive emotion induction improves cardiovascular coping with a cognitive task"

_PeerJ, doi:10.7717/peerj.10904_

## Round 0.1 · original submission · Major Revisions

Dear Dr. Serrano-Rosa,

First let me apologize for delaying in answering due to the hardship arising from the COVID-19 pandemic.

Please find below the comments from the two reviewers.

With regard to the slightly divergent recommendations on the Introduction, I would suggest keeping it thorough and enriching it on the basis of the suggestions made by reviewer 1.

I would not use the label 'sex', but rather 'gender'.

I am looking forward to your revision.

With kind regards,
Claudia Scorolli

Reviewer 1 ·

Basic reporting

The introduction is well written however needs more detail. I suggest that the authors improve the description of the following aspect:

No mention regarding how the author intend to induce positive emotional state. They refer to it in the Method section, but I think that it is important to mention it in the introduction too, to give the reader the idea of what they are going to manipulate and how.

Line 72 – please elaborate better the link between the prefrontal cortex activation – positive emotion – cognitive skills.

Line 95 - considering heart rate variability (HRV) as an alternative to neuroimaging tecniques seems a bit risky. Rephrase it.

The hypothesis need to be more specific:
For example: specify what do you intend with higher performance in the Tower of Hanoi.
The SAM estimate three dimensions: valence, activation and dominance, why authors have hypothesis only on the valence dimension?

Regarding article structure: I would suggest to move the protocol before statistical analysis, after the description of all variable that are going to be tested. The procedure will be clearer to the reader.

Experimental design

The research question is well defined, but the investigation chosen by the authors, employing the Tower of Hanoi to evaluate cognitive skill is not sufficient. The authors, however, report already this as a limitation of the study.

How was calculated the sample size? This information is missing.

Validity of the findings

no comment

Additional comments

Minor comment:
line 266: sexes is sex
pag. 25 table 1 is table 2.
Pag. 27 table 1 is table 3.
ZEMG results would be easier reported in a Table.

·

Basic reporting

The introduction part could be more concise. Some of the information is distracting and redundant.

Experimental design

L189. Please elaborate more on the details if students know they will get positive/negative intervention beforehand. Does the observer know which one is going to give the intervention? Is the study double-blinded or triple-blinded? These will affect the results.

Validity of the findings

Findings are impressive. However, the sample size and selection might be an issue that has already been discussed in the discussion part. Observer bias based on the prior studies/beliefs could exist, and readers may like to see more comments on this part. The sample selection could make these findings unconvincing as well.

---

## Round 0.2 · accepted · Accept

Dear Dr. Serrano,
Thank you for your revisions that improved the manuscript.
Since Reviewer 1, who is no longer available, had suggested introducing Table 4 (for ZEMG results), I would keep it.
Thank you,
Kind regards,
Claudia Scorolli

·

Basic reporting

Well written

Experimental design

Already revised the methodology based on the previous suggestions. Appreciate the clarification of how the investigators and participants were involved.

Validity of the findings

well written